# Hierarchical Graph Network for Fullerene Energy Prediction

Tong Cai[1][0009−0002−7268−1717], Asiri Wijesinghe[2][0000−0003−4392−5348], Sergio J. Rodríguez Méndez[1][0000−0001−7203−8399], and Amanda J. Parker[1][0000−0003−2207−744X]

[1] The Australian National University, Canberra ACT 2601, AU
carina.cai@anu.edu.au
[2] CSIRO's Data61, Canberra, ACT 2601, AU

## 1 Introduction

Fullerenes are cage-like carbon allotropes composed of atoms arranged in closed polyhedral geometries with a combination of pentagonal and hexagonal faces. This structure gives rise to diverse molecules (with $C_{60}$ being the most iconic) and applications in photovoltaics, energy storage, drug delivery and phototherapy [9, 2]. Predicting the quantum properties of fullerenes is therefore crucial for accelerating carbon-based materials discovery and biomedical applications.

While graph neural networks (GNNs) have achieved strong performance in molecular property prediction [10, 5, 4], fullerenes pose unique challenges: uniform $sp^2$ hybridisation creates nearly identical local atomic environments, and standard message passing neural networks (MPNNs) overlook their multi-scale cage geometry. Informed by observed structure-property dependencies [6], we propose a hierarchical message passing network that models atom-, face-, and cage-level interactions, leveraging 5- and 6-fold symmetries for more expressive and symmetry-aware representations. Our approach outperforms standard MPNNs on fullerene energy prediction, providing a principled geometric framework for symmetric molecular systems that has potential for generalisation to larger or more diverse nanoparticle, lattice, or carbon allotrope systems.

## 2 Method

**Notations and Preliminaries:** Let $\mathcal{G}_{\text{atom}} = (\mathcal{V}_{\text{atom}}, \mathcal{E}_{\text{atom}})$ denote the atom-level graph, where each node $v_i \in \mathcal{V}_{\text{atom}}$ represents a carbon atom (with spatial coordinates $\mathbf{x}_i \in \mathbb{R}^3$), and edges correspond to covalent bonds forming a 3-regular graph. Let $\mathcal{G}_{\text{face}} = (\mathcal{V}_{\text{face}}, \mathcal{E}_{\text{face}})$ denote the dual face-level graph, where each node $f_k \in \mathcal{V}_{\text{face}}$ represents a pentagonal or hexagonal face, with edges defined by shared bonds in $\mathcal{G}_{\text{atom}}$. By Euler's polyhedral theorem, all fullerenes contain exactly 12 pentagons: $\mathcal{V}_{\text{face}} = \mathcal{F}_{\text{pen}} \cup \mathcal{F}_{\text{hex}}$, with $|\mathcal{F}_{\text{pen}}| = 12$. Formal definitions of vertices, edges, coordinates, and associated features are summarised in Table 2 (Appendix A). Fig. 1 illustrates the atom–face dual hierarchy.

**Model Details** Our hierarchical message passing architecture consists of $L$

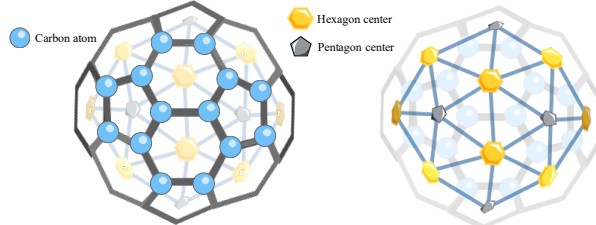

**Fig. 1.** Highlighted Atom-level graph (left) and face-level dual graph (right).

layers, each performing three sequential operations that respect fullerene's multi-scale structure. Fig. 2 provides an overview.

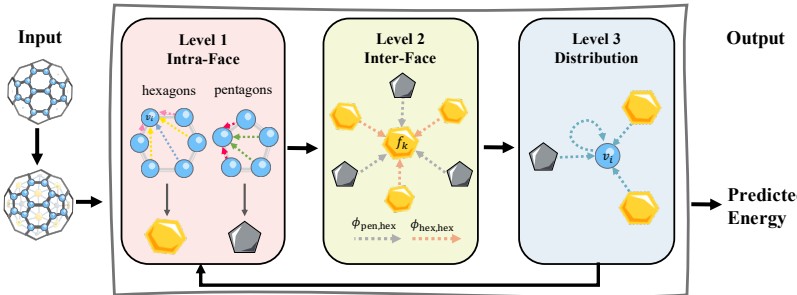

**Fig. 2.** Hierarchical message passing scheme. Different message passing networks are denoted using dashed lines of different colours.

**Level 1: Intra-Face Message Passing.** We model ring structures on the fullerene surface using hop-based message passing that respects cyclic symmetry. Separate networks for pentagons ($C_5$-MPNN) and hexagons ($C_6$-MPNN) use hop-specific parameters to capture intra-ring interactions. This enables learning of distinct patterns based on ring geometry in a single step.

At level $l$, atom features are summed to pool the corresponding face-level representations $\mathbf{h}_{f_k}$, where $\mathcal{A}(f_k)$ denotes the set of atoms belonging to face $f_k$:

$$\mathbf{h}_{f_k}^{(l)} = \text{POOL}\left(\mathbf{h}_{v_i}^{(l)} \mid v_i \in \mathcal{A}(f_k)\right) \tag{1}$$

**Level 2: Global Face-to-Face Processing.** We capture cage-level interactions through global face-to-face message passing. E(3)-invariant geometric features-centroid distances and normal-vector angles-are computed between adjacent faces and expanded via radial basis functions. Three specialised networks handle distinct face-pair types (pentagon–pentagon, pentagon–hexagon, hexagon–hexagon), allowing type-specific interaction learning. Face embeddings are then updated by aggregating messages from neighbouring faces.

**Level 3: Face-to-Atom Distribution.** Each atom belongs to multiple faces. Face-level information is propagated back to atoms, combining local and global structural context, for richer atomic representations.

$$\mathbf{h}_{v_i}^{(l+1)} = \text{COMBINE}\left(\mathbf{h}_{v_i}^{(l)}, \bigoplus_{f_k:v_i\in\mathcal{A}(f_k)} \mathbf{h}_{f_k}^{(l)}\right) \tag{2}$$

**Energy Prediction.** After $L$ layers, atom features capture multi-scale geometric information. A final molecular energy prediction is obtained by aggregating atomic representations and applying a MLP.

## 3 Experiments

We evaluate on the 2,487 neutral fullerenes from the Fullerene dataset [1], which includes Density functional tight binding (DFTB) relaxed structures and 7 quantum properties. We focus on distortion energy, band gap, and Fermi energy ($E_F$) due to their relevance to stability and electronic behaviour.

We implement a 2-layer model with hidden dimension 128, using one intra-face and three inter-face message passing steps per layer. Training uses AdamW [7] (learning rate $10^{-4}$, weight decay $10^{-5}$), batch size 64, and early stopping over 100 epochs (patience 10). Gradient clipping (max norm 1.0) is applied. The model has 1–2M parameters, and the best checkpoint is selected based on validation performance.

**Baselines:** (1a-b) 66 domain expert-defined global geometric features fit with classical ML models - Random Forest (RF), artificial neural network (ANN); (2a-d) standard MPNNs using atom- and face-level graphs with geometric inputs (distance and angles) and (3) MEGNet [3], which represents state-of-the-art molecule models which are also tractable on larger fullerenes.

**Table 1.** Model comparison for fullerene energy predictions.

| Model | Input | Distortion E | | $E_F$ | | Band Gap | |
|---|---|---|---|---|---|---|---|
| | | MAE | $R^2$ | MAE | $R^2$ | MAE | $R^2$ |
| (1a) Random Forest | Expert Tabular | 0.5187 | 0.95 | 0.0595 | 0.60 | 0.1334 | 0.86 |
| (1b) ANN | Expert Tabular | 1.5916 | 0.66 | 0.1490 | $-1.14$ | 0.2363 | 0.60 |
| (2a) MPNN Distance | Atom Graph | 1.4212 | 0.69 | 0.0839 | 0.22 | 0.0461 | 0.68 |
| (2b) MPNN Distance | Face Graph | 2.1487 | 0.30 | 0.0848 | 0.19 | 0.0453 | 0.70 |
| (2c) MPNN Angular | Atom Graph | 2.0507 | 0.35 | 0.0843 | 0.20 | 0.0465 | 0.69 |
| (2d) MPNN Angular | Face Graph | 1.9998 | 0.37 | 0.0858 | 0.19 | 0.0488 | 0.69 |
| (3) MEGNet | Atom Graph | 2.8714 | $-0.08$ | 0.0934 | 0.00 | 0.4087 | -0.04 |
| **Ours** | **Atom+Face** | 0.9303 | 0.82 | 0.0791 | 0.37 | 0.1614 | 0.79 |

**Results:** Table 1 reports mean average error (MAE) and $R^2$ scores, while Fig. 3 shows predicted vs. ground truth values. Our model outperforms all benchmarks except for the domain expert-defined features. The improvement over standard

atom-level MPNNs is achieved by capturing two key aspects: (1) ring symmetries via hop-based intra-ring message passing; and (2) cage-level interactions through face-to-face updates.

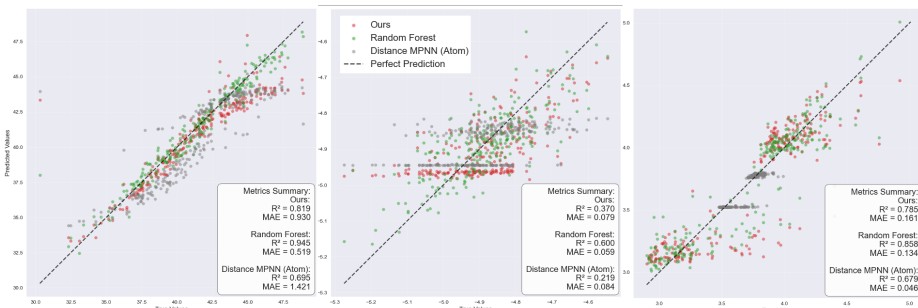

**Fig. 3.** Predicted energies vs. ground truth. distortion energy (left), Fermi energy $E_F$ (center), and band gap (right).

While our hierarchical GNN captures structural patterns critical for distortion and band gap prediction, it struggles with low range values of $E_F$ (as does the MPNN), this is likely due to this property's dependence on global electronic structure. Some longer-range or global structural detail may be necessary for accurate $E_F$ prediction, or incorporating electronic structure representations, such as orbital information or long-range electronic correlations. Notably, RF models using hand-crafted global features is the strongest performing baseline, highlighting the effectiveness of domain-informed descriptors. Our method offers a geometry-aware alternative with better interpretability and generalisability for symmetric molecular systems or repetitive lattice structures like fullerenes.

## 4   Conclusion

We propose a hierarchical message passing network exploiting fullerene's multi-scale structure that substantially outperforms standard MPNNs on 2,487 fullerene structures. While hand-crafted features with RF remain the benchmark on accuracy, our work demonstrates that hierarchical architectures respecting $C_5$ and $C_6$ ring symmetries are fundamentally better suited to highly symmetric molecular systems-and crucially, our method is generalisable across polyhedral structures without expensive quantum chemical pre-computation. This has direct implications for drug delivery and biomedical applications: efficient computational property prediction on large fullerenes enables rapid virtual screening of biocompatible candidates. This is critical for designing functionalised fullerenes that are promising therapeutic carriers with tunable optoelectronic and delivery properties for drug delivery, phototherapy, and diagnostics [2, 8]. This work establishes principles for designing symmetry-aware GNNs applicable to carbon nanotubes, nanoparticles, and other biologically relevant polyhedral systems, opening new possibilities for materials discovery in medicine.

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

# Appendices

# A   Dual-level Representation

**Table 2.** Graph representations for fullerenes

| Component | Atom-level | Face-level |
|---|---|---|
| **Vertices** | $\{v_i\}_{i=1}^{N_{\text{atom}}}$ | $\mathcal{F}_{\text{pen}} \cup \mathcal{F}_{\text{hex}}$ ($|\mathcal{F}_{\text{pen}}| = 12$) |
| **Edges** | Covalent bonds | Shared edges in $\mathcal{G}_{\text{atom}}$ |
| **Positions** | $\mathbf{x}_i \in \mathbb{R}^3$ (atomic coord.) | $\mathbf{c}_k \in \mathbb{R}^3$ (face centers) |
| **Node Features** | Radial distance from origin Pentagon/hexagon membership | Face type, area, radial distance from origin |
| **Property** | 3-regular | $|\mathcal{V}_{\text{face}}| - |\mathcal{E}_{\text{face}}| + |\mathcal{V}_{\text{atom}}| = 2$ |

## B    Computational Efficiency and Practical Advantages

### B.1    Training and Inference Time Complexity

Our hierarchical architecture exhibits linear time complexity per layer. Each layer performs three sequential operations:

- Intra-face message passing: $\mathcal{O}(|\mathcal{V}_{\text{face}}|)$
- Inter-face message passing: $\mathcal{O}(|\mathcal{V}_{\text{face}}|)$ with fixed message-passing steps (or $\mathcal{O}(|\mathcal{V}_{\text{face}}|^2)$ if a fully-connected dual graph is considered
- Face-to-atom distribution: $\mathcal{O}(|\mathcal{V}_{\text{atom}}|)$

Given the fullerene topology constrains that the structures are made up of 12 pentagons and $(|\mathcal{V}_{\text{atom}}|/2 - 10)$ hexagons, the overall complexity per layer is $\mathcal{O}(|\mathcal{V}_{\text{atom}}|)$, linear in molecular size.

Empirically, on a single GPU (NVIDIA RTX 4090), training takes approximately 10 minutes per epoch on the 2,487 fullerenes using the above hyperparameter setting. These times are competitive with standard 2-layer MPNN baselines.

### B.2    Feature Extraction Cost and Practical Advantages

A critical practical distinction emerges when accounting for feature preparation costs—an often-overlooked consideration in molecular machine learning pipelines.

**Tabular Baseline:** The 66 hand-engineered features (extracted from a pool of 830 features) underlying the RF baseline are not "free"—they require extensive domain expertise and specialised computation.

**Our Hierarchical GNN:**  Face and atom initial features are pre-computed during graph construction with linear complexity in molecular size, dominated by adjacency matrix construction. On average, it takes 0.1 seconds per molecule in the dataset.

### B.3    Generalisation Capability and Transferability

Beyond computational accuracy and efficiency, our approach offers a critical practical advantage: generalisability across polyhedral structures. The hierarchical architecture naturally transfers to unexplored carbon topologies (larger fullerenes, carbon nanotubes, Goldberg polyhedra) without structural redesign or re-training on domain-specific descriptors.

In contrast, the tabular features are fullerene-specific and cannot be transferred to other cage structures without extensive re-engineering of feature definitions and potentially additional quantum chemical calculations. For materials discovery pipelines exploring diverse structural classes, this transferability eliminates the need for bespoke feature engineering at each new system, which is a significant advantage for rapid materials screening in biomedical and nanotechnology applications.

## C    Remaining challenges and future work

### C.1    Addressing Fermi Energy Prediction

GNN-based methods require explicit electronic structure information to predict Fermi energy accurately. Future work should incorporate: (1) orbital-level features from quantum mechanical calculations, (2) long-range electronic interactions beyond geometric adjacency, and (3) mechanisms to capture global charge distributions. Additionally, analysing whether certain fullerene sizes are under-represented in training may reveal data limitations versus architectural constraints.

### C.2    Geometric Enhancements

Incorporating higher-order geometric information could improve performance: (1) surface curvatures and local geometric descriptors, (2) explicit point group symmetries through group-theoretic message passing, and (3) long-range pentagon clustering effects—current model captures only adjacent face interactions, but global pentagon distributions also influence cage stability.

### C.3    Architecture Improvements

Equivariant architectures (SE(3) or E(3)-based) may better preserve 3D constraints. Attention mechanisms learning which geometric features matter for each property could improve flexibility beyond hand-designed hop distances. Hybrid models combining hierarchical message passing with learned electronic descriptors may bridge geometric and electronic reasoning.

### C.4    Generalisation

Extending to larger fullerenes, charge species, and other polyhedral molecules (carbon nanotubes, MOFs) would assess generalisation.

### C.5    Inverse Design Applications

Forward property prediction may insufficiently test hierarchical structure learning. Inverse design that generates structures with target properties inherently requires simultaneous reasoning about local (ring stability) and global (pentagon distribution) constraints, and would likely demonstrate hierarchical inductive biases more convincingly.