# OpenReview forum: "Hierarchical Graph Network for Fullerene Energy Prediction"
_AJCAI/2025/Workshop/AIML-CEB — AIML-CEB 2025 Oral_

### Official Review · Reviewer_NwT8 · 2025-11-04
**Interesting idea, well written. Lacks statements of biological relevance, but otherwise a nice paper.**

**Rating:** 7
**Confidence:** 4

**Review:**

The paper presents an original GNN design that utilises a hierarchical message passing system tailored towards the geometric structures of fullerene. The model outperforms baseline deep learning methods, but not RF with handcrafted features.

pros:

1. The proposed hierarchical message passing seems interesting, well designed, and convincingly suitable for the geometric structures of interest, evident by outperforming SOTA models on a benchmark dataset. The author(s) displayed expertise in both computational and materials domains, and a clear understanding of the research problem, the method, and the limitations.

2. The papers reads well. The figures are appropriate and clearly explain a complex system.

3. While the proposed method is outperformed by handcrafted features + RF,  the author(s) was able to identify potential causes and pose interesting future directions to address the problem.

cons:

1. While the submission tackles molecular modelling in a way that could benefit research on biomolecules, the subject of analysis (fullerene) isn't usually associated with the biological sciences. The author(s) have pointed out potential applications and significance of research in materials and the physical sciences, but not in biologically-relevant fields.

2. While the paper clearly explains the conceptual structure of the proposed GNN, specific architectural details are a bit blurry, which can cause a bit of confusion.  e.g. in equation 1, the pooling operation POOL is unspecified, and the notation h (hidden representation?) is undefined. However, one could say that this is understandable due to the page limit.

---

### Official Review · Reviewer_4GSZ · 2025-11-06
**GNNs for Fullerene energy prediction**

**Rating:** 8
**Confidence:** 4

**Review:**

This is a lovely piece of work that introduced an interesting new domain-specific GNN architecture and well executed. The description of the method is clear, and the experiment is interesting, with a thorough analysis.

It's interesting how the random forest + domain specific features is such a strong baseline, and it would have been nice to have more description of why the authors think this is the case. I wonder if you have tried other stronger ensemble methods, such as gradient boosted  trees, and if these methods perform better or worse? I also wonder if the GNNs/ANNs are over or under fitting, explaining their worse performance compared to the RF (where randomisation tends to be a decent regularisation strategy). Where training folds are fixed (such as your 80:10:10 splits), human-in-the-loop overfitting can still happen where you select architectures/hyper-parameters on the validation set, essentially overfitting to it. If you haven't already, I would suggest resampling these folds, while holding the model parameters fixed, and then seeing if validation performance suddenly drops. Alternatively, it could be a NN architectural issue, e.g. sometimes RELU's can lead to "dead neurons" (maybe explaining the middle plot in figure 3), and using LeakyRELU or GELU etc can help (along with residual/skip layers and layer/batch normalisation where appropriate).

Well done, this was an enjoyable read!

---

### Official Review · Reviewer_FAWa · 2025-11-09
**Optimizing GNN for fullerene energy predicition**

**Rating:** 10
**Confidence:** 4

**Review:**

This paper proposes a hierarchical message-passing network for fullerene energy prediction, with three levels—Intra-Face, Inter-Face, and Face-to-Atom Distribution—to capture ring symmetries and cage-level interactions. The approach outperforms standard MPNNs on the fullerene dataset, though a Random Forest with hand-crafted global features remains the strongest overall baseline.

Pros

•	Clear baseline setup and comprehensive performance evaluation.
•	Honest discussion of limitations (e.g., Fermi energy).

Cons

•	Add comparisons on efficiency (training/inference time) and feature-extraction cost to better showcase practical advantages over tabular baselines and standard MPNNs.

---

### Decision · Program_Chairs · 2025-11-12

Accept (Oral)